# Fermentation Characteristics and Aromatic Profiles of Plum Wines Produced with *Hanseniaspora thailandica* Zal1 and Common Wine Yeasts

**DOI:** 10.3390/molecules28073009

**Published:** 2023-03-28

**Authors:** Nanthavut Niyomvong, Chanaporn Trakunjae, Antika Boondaeng

**Affiliations:** 1Department of Biology and Biotechnology, Faculty of Science and Technology, Nakhon Sawan Rajabhat University, Nakhon Sawan 60000, Thailand; nanthavut.ni@nsru.ac.th; 2Science Center, Nakhon Sawan Rajabhat University, Nakhon Sawan 60000, Thailand; 3Kasetsart Agricultural and Agro-Industrial Product Improvement Institute, Kasetsart University, Bangkok 10900, Thailand; aapcpt@ku.ac.th

**Keywords:** plum wine, *Saccharomyces cerevisiae* var. *burgundy*, *Hanseniaspora thailandica* Zal1, Lalvin EC1118, total phenolic (TP) content, antioxidant activity

## Abstract

Plum has long been cultivated in northern Thailand and evolved into products having long shelf lives. In this study, plum processing was analyzed by comparing the production of plum wine using three types of yeast, *Saccharomyces cerevisiae* var. *burgundy*, *Hanseniaspora thailandica* Zal1, and *S. cerevisiae* Lalvin EC1118. EC1118 exhibited the highest alcohol content (9.31%), similar to that of burgundy (9.21%), and *H. thailandica* Zal1 had the lowest alcohol content (8.07%) after 14 days of fermentation. Plum wine fermented by *S. cerevisiae* var. *burgundy* had the highest total phenolic (TP) content and antioxidant activity of 469.84 ± 6.95 mg GAE/L and 304.36 ± 6.24 µg TE/g, respectively, similar to that fermented by EC1118 (418.27 ± 3.40 mg GAE/L 288.2 ± 7.9 µg TE/g). *H. thailandica* Zal1 exhibited the least amount of TP content and antioxidant activity; however, the volatility produced by *H. thailandica* Zal1 resulted in a plum wine with a distinct aroma.

## 1. Introduction

*Prunus mume*, the scientific name of the Japanese apricot, is a cold weather fruit of the genus *Rosaceae,* which also includes peaches, persimmons, and plums. When the plum is unripe, it has a green color, whereas when it is fully ripe, it turns yellow. The fruit is approximately 2.5 cm in diameter and is typically cultivated in China, Taiwan, and Japan [1]. Plums have a sour and bitter taste, which is unpleasant; therefore, they are not eaten fresh. However, plums have a unique aroma and can be processed into various foods. Plums contain vitamins C and A, which prevent cancer and heart diseases, help mitigate diarrhea, stimulate appetite, aid in digestion, reduce thirst and keep the throat moist, reduce sweating, improve blood flow, prevent osteoporosis, and contain antioxidants [2,3,4].

In Thailand, plums were planted for over 40 years in the Chiang Rai Province, which was initiated by the Royal Project Foundation’s planting promotion. According to a survey in 1991, farmers were encouraged to grow plums in 1987 as an economic crop, which resulted in a total yield comprising 123,608 plum trees [5]. The types of plums cultivated in Thailand are diverse, but they can be divided into two groups based on the source: the Chiang Rai plum and Taiwan plum. Taiwanese plum varieties that produce high-quality fruit are particularly popular in the current market. The main cultivation areas are the Huai Nam Khun Royal Project Development Center, Huai Nam Rin, Mae Poon Luang, Wat Chan, and Ang Khang, Chiang Rai Province [6]. Because plums were cultivated in Thailand for a long time, the overall yield each year exceeds the market demand; therefore, plums are processed into various products that can be stored for a long time using preservation processes. However, plum processing in Thailand does not vary, and most of the plums are pickled or frozen. The quality of fresh plums in Thailand still necessitates a process to add value by processing them into products that can be stored for an extended period of time while maintaining nutritional value.

Plums have long been used as raw materials for alcoholic beverages in Asian countries. The typical process involves steeping the fruits with liquor to achieve diffusion of various substances from the fruit directly into the liquor without microbial fermentation [7]. Plum liquor is popular in Japan and Korea. Generally, people in Asia tend to drink plum liquor in the form of sweetened liquor that is infused with plums and a high amount of sugar is added to the white liquor. This results in a taste that is easier to drink than regular liquor. In Japan, plum liquor is known as Umeshu, which is made by soaking fresh plums with rock sugar in clear liquor and leaving it until the diffusion results in a plum-flavored product. Korea employs the same process to make wine. Plum liquor is called maesil-ju and méijiǔ in China. Taiwan employs different production process that involves two types of plum beverages, méijiǔ which uses *P. mume* plums and lijiu made from *P. salicina* plum, mixed with oolong tea infused liquor, which is called wumeijiu (smoked plum liquor). However, the aforementioned plum wines are not fermented using yeast, which results in low levels of nutritional substances, such as antioxidants or volatile substances that result in good aromas during the fermentation process. Additionally, because of the large amount of added sugar during the production process, the drink is considered a high-energy food and consuming large amounts of such beverages can increase the risk of health problems.

*H. thailandica* is a yeast species that was first described in 2009 by Jindamorakot et al. [8]. Since its discovery, there was limited research on this particular yeast species. However, several studies investigated the potential applications of *H. thailandica* in the food and beverage industry. Maheswari et al. [9] evaluated the potential of *H. thailandica* as a starter culture for wine fermentation. The researchers found that the yeast was able to produce high levels of desirable aroma compounds and had a strong ability to ferment fructose, making it a potentially useful strain for wine production.

In this study, winemaking using fresh plums from the Royal Project Foundation was investigated using different yeasts, including *Saccharomyces cerevisiae* var. *burgundy*, *H. thailandica* Zal1, and the commercial yeast *S. cerevisiae* Lalvin EC1118. Consequently, the differences in the chemical properties and quality of each plum wine were determined. This was especially the case for *H. thailandica* Zal1, which was isolated from salak fruit and gives a unique aroma of salak fruit with a pleasant fruity scent. Although there were previous studies, a more thorough investigation of the fermentation process and the examination of the compounds that contribute to the unique aroma produced during fermentation, which is a highlight of the particular yeast strain, has not been conducted. Therefore, this study was conducted to explore and investigate these issues. This study aims to focus on biochemicals profiles from fermentation process and investigate the quality of wine produced by *H. thailandica* Zal1 compare with the conventional wine yeast. This research may provide information for using non-*Saccharomyces* yeast to create a unique aroma for wine production. The trend of low-alcohol wine popularity in Thailand requires winemakers to produce wine within a shorter fermentation period to maintain the characteristic fruit flavor and aroma. This research could increase the value of plums grown in the country and lead to an increase in agricultural production value through the application of fermentation science.

## 2. Results and Discussion

### 2.1. Chemical Characteristics of Plums

The general chemical characteristics of ripened plums were measured. On average, the TSS and reducing sugar content of the plums were 9.77 °Bx and 7.19%, respectively. The pH and TTA were 2.61 and 2.47%, respectively. When the chemical compositions of plums were compared with those of previously reported values, the TSS values ranged from 9.5 to 10.0 °Bx, pH ranged from 2.5 to 2.7, TTAs ranged from 2.0 to 3.8% in mature-stage fruits and 4.0–5.7% in green-stage fruits [10], and the reducing sugar content ranged from 5.18 to 8.56% [11], which was similar to the results of the current experiment.

### 2.2. Fermentation Profiles of Plum Wine Using Different Yeasts

The fermentation of plum wine from the three yeast species was studied from samples at 28 °C for 14 d. The samples were taken every 24 h and the TTA (% of citric acid), pH, TSS (°Bx), and alcohol content were measured. The results of all three wine samples portrayed similar results. During the plum wine fermentation process, the initial acid contents of the plum wines were relatively high; the acid content of the samples was similar, ranging from 1.072 to 1.120%. The final acid contents of the plum wines increased slightly, ranging from 1.264 to 1.392% (Figure 1a). Therefore, the pH in all experiments decreased from 3.0 to 2.42–2.52 because of the plums’ high acidity. In addition, the total acid contents in all treatments also increased during the plum wine fermentation process because yeast growth produces large amounts of organic acids. Non-volatile acids comprise the total acid content. The primary non-volatile acids include tartaric, citric, and malic acids, which are crucial ingredients in wine because of their influence on taste and color [12]. Malic and citric acids are reused for intracellular metabolism using yeast, and tartaric acid precipitates in the form of potassium bitartrate (cream of tartar), which reduces the specific gravity of wine, and precipitation occurs when the wine is stored at low temperatures. After fermentation, non-volatile acids inhibit the growth of volatile acid-producing microorganisms [13]. Each plum wine’s volatile acid content represented by acetic acid increased from the first to last day, when the final volatile acid content was 0.0023–0.0024% (Figure 1b). The maximum acceptable limit for VA represented in acetic acid content in most wine is 1.2 g/L [14]. The amount of acetic acid in wine should be minimal, as large quantities indicate a deterioration in wine quality caused by the acetic acid-producing bacteria, particularly with regard to *Acetobacter acetii*. However, after fermentation, volatile acidity increases owing to bacterial activity or alcohol oxidation [15].

Figure 2 shows that the reducing sugar content in wines fermented by *S. cerevisiae* var. *burgundy*, *H. thailandica* Zal1, and EC1118 were highest on the seventh, fourth, and fourth days, respectively. The hydrolysis of sucrose generated a mixture of fructose and glucose by the invertase enzyme in yeast, which resulted in an increase in reducing sugars. The reducing sugar is converted to ethanol and carbon dioxide by other enzymes in the Embden–Meyerhof–Panas (EMP) pathway [16]. Subsequently, the reducing sugar was converted to alcohol. The highest alcohol content of plum wine observed was in EC1118 (9.31%), which was similar to that of *S. cerevisiae* var. *burgundy* (9.21%) but higher than that of *H. thailandica* Zal1 (8.07%). Alcohol content was relatively stable during the last days of fermentation. Boonsupa and Kerdchan [17] studied wine fermentation of three Prunus species, *Prunus persica* L., *Prunus domestica* L., and *Prunus mume* L., using *S. cerevisiae*. After five days of fermentation at 28 °C, they found that Chinese plum (*Prunus mume* L.), red plum (*Prunus domestica* L.), and peach (*Prunus persica* L.) had alcohol contents of 6.23%, 6.72%, and 13.81%, respectively. The alcohol content of the Chinese plum wine in this study was higher than that reported by Boonsupa and Kerdchan [17], which may be due to the longer fermentation time.

Figure 3 shows that the yeast populations from all treatments continued to increase in the first stage and, subsequently, decreased slightly. The fermentation patterns of the *S. cerevisiae* var. *burgundy* and *H. thailandica* Zal1 on the third day demonstrated that the yeast population increased by approximately two log cycles, whereas the yeast population of commercial yeast EC1118 increased by approximately two log cycles on the first day. In the first stage of fermentation, the fermentable sugar content in fruit juice was sufficient for yeast growth until the log phase. Subsequently, the yeast population of *S. cerevisiae* var. *burgundy* and *H. thailandica* Zal1 continuously decreased by approximately four log cycles through the last day, and the commercial yeast EC1118 decreased by three log cycles. Increasing alcohol levels during plum wine fermentation inhibits yeast growth and reduces growth, fermentation rate, and cell viability [18].

### 2.3. Total Phenolic Content and Antioxidant Activity

Generally, TP analysis measures the reduction capacity of the sample in a similar way to antioxidant activity analysis. Several studies reported a linear relationship between TP and antioxidant activity [19]. Similarly, the TP content of plum wine fermented by *S. cerevisiae* var. *burgundy* and commercial yeast EC118 increased from the first to last day, which is similar to the trend of antioxidant activity. The highest phenolic content was determined by the Folin–Ciocalteu reagent, and total antioxidant activities were determined by DPPH of plum wine fermented by *S. cerevisiae* var. *burgundy* were 469.84 ± 6.95 mg/L of gallic acid equivalents (GAE) and 304.36 ± 6.24 µg/L of Trolox equivalents (TE), respectively (Table 1), which was significantly different with plum wine fermented by *H. thailandica* Zal1 and EC1118. The TP content and antioxidant activity of plum wine fermented by *H. thailandica* Zal1 decreased slightly from the first to the last day. This result was in agreement with the study of Towantakavanit et al. [20], which reported that the decrease in the TP level could have been due to modification of the phenolic composition by processes such as oxidation, condensation, and polymerization. The obtained results were consistent with those of these studies. The TP contents of Chinese plum, red plum, and peach wines produced by the wine yeast *S. cerevisiae* were 66.39, 215.85, and 140.51 mg GAE/L, respectively, as reported by Boonsupa and Kerdchan [17]. Miljić et al. [21] also reported the TP content of plum wine (*Prunus domestica* L.) ranged from 870 to 1160 mg GAE/L using the commercial wine yeast *S. cerevisiae.* The antioxidant activity of plum wine in this study was similar to foreign red and white wines; specifically, plum wine indicated the same level of antioxidant activity as white wine which ranged from 152.7 to 445.52 µg TE/g [22]. Moreover, the TP content and antioxidant activity in plum wine depend on several factors, including variety and the winemaking process [21,23,24].

### 2.4. Wine Aroma Profiles by GC-MS

Preliminary chemical composition analysis of the plum wines using different yeast inocula by GC-MS revealed that the plum wines fermented by the yeasts in this study contained three identical volatile substances: ethanol, acetic acid ethyl ester, and isopentyl alcohol (Table 2). *S. cerevisiae* var. *burgundy* produced two additional volatile substances, isobutylalcohol and 2,3-Butanediol, whereas *H. thailandica* Zal1 produced two more volatile substances in ester groups, namely propanoic acid, 2-methyl-, ethyl ester, and 1-Butanol, 3-methyl-, acetate. For commercial yeast, EC1118 produced fewer volatile substances than the other two strains. A previous study reported that the production of esters during alcoholic fermentation was similarly carried out in different types of yeast, and that acetate esters depended on the balance between the synthesis and hydrolysis of esters by acetyl alcohol transferase and ester-hydrases [25]. Although *S. cerevisiae* is the most common ingredient in wine, other genera and species of yeast known as non-*Saccharomyces* are also present, which are found in grapes, must, and wines and enhances the flavor of the wine [26]. Some researchers reported that wine aroma is determined by the metabolism of *S. cerevisiae*, while others believe that wine aroma is determined by non-*Saccharomyces* [27,28], which produces high concentrations of fermentation compounds (acetic acid, glycerol, acetoin, acetate esters, etc.) [25]. Tristezza et al. [29] studied wine fermentation using mixed cultures of *S.cerevisiae* and *H. uvarum* ITEM8795. The mixed culture could enhance the organoleptic quality of wine and simultaneously reduce volatile acidity. *Hanseniaspora* spp. is essential in the production of volatile compounds in wine and its general chemical composition of wine made from mixed cultures *Hanseniaspora* spp./*S. cerevisiae* differs from the composition of wines produced from a single *S. cerevisiae* [30,31,32,33]. 

### 2.5. FTIR Analysis

FTIR was used to analyze the presence of TP and antioxidant compounds in plum wines. The FTIR peaks displayed functional group oscillations corresponding to the various biomolecules. Figure 4 shows the spectra of each plum wine in the range 4000–500 /cm.

The spectral bands at 1500–400/cm are called fingerprints and are used to differentiate between food types [34]. When plum wines were fermented by *S. cerevisiae* var. *burgundy* and *H. thailandica* Zal1, the vibration patterns of the spectra in the range 1500–400/cm were similar. In contrast, the spectral vibrating pattern of plum wine fermented by commercial yeast EC1118 is different from the other plum wines listed above, with the possibility of producing substances other than plum wines. The main components of these plum wines are consistent with those reported in previous studies. Wavelengths of 3800–2790/cm were reported to be the oscillating range of the –OH group of water and the C–H stretching of acetic acid. The absorption at 1300–1000/cm corresponds to the C–O stretching functional group of the organic acids, whereas the bands located at 1100–1000/cm are due to the C–O stretching of ethanol. The peaks found in the range of 1700–1600/cm indicate the C=O stretching functional group in the aldehyde compound’s structure, whereas the wavelength range of 1800–900/cm indicates the –C–O and –OH groups in phenolic compounds [35,36]. This is due to the presence of phenolic acids and confirmation of antioxidant activity in plum wines.

The antioxidant ability in wine is related to TP content because phenolic compounds are a substance group with antioxidant properties. The mechanism of antioxidant activity involves the supply of hydrogen atoms to free radicals, which terminates the chain reaction. The use of DPPH radicals is a model for the antioxidant activity of a substance, which can be performed faster than other methods. The antioxidant activity of wine against DPPH radicals is caused by the presence of hydrogen atoms in the wine [37], and when DPPH radicals gain electrons, it will be in a stable condition [38]. However, TP content and antioxidant capacity depend on the type of fruit and wine being processed.

This result may be used as an alternative approach to solve the problem of plum production oversupply and improve the value of products simultaneously through the creation of functional beverages. Additionally, winemaking is typically performed using grapes. In Thailand, grapes are expensive to produce due to their high care costs, resulting in high production costs. On the other hand, plums are less prone to diseases and pests and have good yields in Thailand. It is beneficial to use plums that are readily available and present low-cost raw materials. Specifically, the resulting fermentation activity can trace the formation of various active substances, leading to low-cost, high-quality winemaking, and further boost the production of high-quality wine produced in the area.

## 3. Materials and Methods

### 3.1. Preparation of Staters

Yeast cultures *of S. cerevisiae* var. burgundy and *H. thailandica* Zal1 in Yeast Extract peptone Dextrose Agar (YPD) medium were incubated at 28 °C for 24 h. Next, a full loop of colony growing in YPD medium was transferred into 250 mL Erlenmeyer flask containing 100 mL of sterile plum juice and incubated at 28 °C for 24 h and 5% (*v*/*v*, 10^6^ CFU/mL) was used as inoculum for plum wine fermentation. Commercial wine yeast (*S. cerevisiae* Lalvin EC1118) in a ratio of 0.25 g /L was dissolved in warm water (35–37 °C) and left for 15–20 min before use.

### 3.2. Plum Juice Preparation and Plum Wine Fermentation

The ripe yellow plums obtained from the Royal Project Foundation in northern Thailand were washed, dried, and pedicel was removed with a toothpick. The fruits were crushed by hand and the ratio of fruit to water was adjusted to 1 kg of fruit: 4 L of water. The total soluble sugar (TSS) and pH were adjusted to 25 °Bx and 3–4, respectively, by adding sucrose and baking soda. Potassium metabisulfite (K_2_S_2_O_5_) at a final concentration of 75–100 mg/L was added to the mixture and left overnight for decontamination. A seed culture of 5% (*v*/*v*, 10^6^ CFU/mL) or EC1118 at 0.25 g/L was transferred to the individual mixtures, and fermentation was conducted at 25 °C for 14 d. Samples were collected every 24 h over 14 d and used for yeast population and chemical analysis. For yeast population, the wine sample (10 mL) was placed in a flask with 90 mL of sterile normal saline (0.85%) using aseptic techniques and was mixed thoroughly. The sample was diluted by pipetting 1 mL of the wine sample into 9 mL of sterile saline and suitably diluting the wine sample. Next, 0.1 mL of wine samples at various dilution levels were plated on Plate Count Agar (PCA), spread on an agar plate using a sterile spreader, and left to dry. The plates were incubated at 28 °C for 24–48 h. The total number of yeast colonies in the culture medium was counted and reported as CFU/mL. All experiments were conducted in triplicates.

### 3.3. Chemical Analysis

Total titratable acidity (TTA) [39] and volatile acidity (VA) [40] were analyzed based on citric acid and acetic acid, respectively, via titration with 0.1 N NaOH using phenolphthalein as an indicator. Prior to titration, volatile acids were separated from the wine samples by steam distillation. The pH was measured using a pH meter (PH1200; Horiba, Japan). The TSS was measured at 20 °C using a hand refractometer (RHB-32ATC, Shenzhen City, China), reported as °Bx for soluble solid content. The reducing sugar content was estimated using the Nelson–Somogyi assay [41].

The ethanol concentration was analyzed using gas chromatography (Chromosorb-103, GC4000; GL Sciences; Tokyo, Japan) with an HP5 capillary (30 m × 0.32 mm × 0.25 µm; JW Scientific; Folsom, CA, USA) and FID detector under the following conditions: split flow, 50 mL/min; air flow, 250 mL/min; N2 carrier flow, 30 mL/min; column temperature, 185 °C; injector temperature, 250 °C; detector temperature, 250 °C. n-Propanol was used as the internal standard for comparison [42]. All measurements were performed in triplicate.

### 3.4. Total Phenolic (TP) Analysis

The Folin–Ciocalteu colorimetric method was used to determine the TP compounds [43]. Each sample (0.3 mL) was mixed with 1.5 mL of Folin–Ciocalteu reagent dilution (Sigma-Aldrich, Saint Louis, USA). After incubation in the dark for 5 min, 1.2 mL of 7.5% sodium carbonate solution (Na_2_CO_3_) was added and mixed thoroughly. The mixture was incubated in the dark for 30 min before measuring the absorbance by a spectrophotometer (Thermo Fisher Scientific 4001/4 Genesys 20, Waltham, MA, USA) at a 765 nm wavelength. A calibration curve was established using gallic acid standard solution.

### 3.5. Antioxidant Activity Analysis

The total antioxidant activity of the wine samples was determined using the DPPH Radical Scavenging Capacity Assay described by Vidal-Gutiérrez et al. [44]. Briefly, 1 mL of each sample was mixed with 0.3 mL of DPPH solution and stored in the dark for 30 min, and the absorbance was measured by spectrophotometry at a wavelength of 517 nm with a UV-vis spectrophotometer (Shimadzu model UVmini-1240, Kyoto, Japan). The results were expressed in µg Trolox equivalent (µg TE/g) of the wine sample.

### 3.6. Fourier Transform Infrared (FTIR) Analysis

FTIR (Nicolet IR200 FTIR, Thermo Scientific, Madison, WI, USA) was used to identify the characteristic functional groups in the sample. The spectra were recorded at a range of 500–4000/cm with a mean of 32 scans and a resolution of 4/cm. The FTIR spectra were plotted as intensity versus wave number [45].

### 3.7. Gas Chromatography-Mass Spectrometry (GC-MS) Analysis

The volatile compounds in the samples were quantified by GC-MS (Shimadzu, Nexis GC-2030NX, Japan) and a DB-5MS column (30 m × 0.32 mm, 0.5 µm; Agilent, Santa Clara, CA, USA) [46]. The following steps were taken to prepare an analytical headspace gas chromatographic sample: 1 g of the wine sample was weighted, placed in a 20 mL headspace vial, and heated to 60 °C for 20 min. Next, the auto injector (HS-20, Shimadzu, Kyoto, Japan) absorbed the contents in the headspace vial area, which was injected into the gas chromatograph to conduct gas mass spectrometry in split mode at 280 °C. Chromatographic separation was conducted using the following method: the column was maintained at 80 °C for 5 min, increased to 150 °C at a rate of 5 °C/min, and subsequently increased to 280 °C at a rate of 10 °C/min. Helium was used as the carrier gas at a constant flow rate of 1.49 mL/min. The GC-MS analysis was performed in EI (70 eV) with an ion source temperature of 280 °C, an interface temperature of 280 °C, and a solvent cut time of 0.5 min. Identification of volatile compounds was performed using spectral comparison with a mass spectral library of Wiley 7 NIST 12 and NIST 62 analyzed in scan mode in the range of 35–500 masses.

### 3.8. Statistical Analysis

Differences between the treatment means were evaluated by statistical analysis of variance followed by Duncan’s multiple range test using SPSS Software (version 20.0; IBM Analytics, New York, NY, USA). The mean values were considered significantly different at *p* < 0.05.

## 4. Conclusions

In conclusion, these results demonstrate that plum wine from each yeast fermentation process had different properties. Using *S. cerevisiae* var. *burgundy*, a common wine yeast, and commercial yeast EC1118, resulted in an increase in TP content and antioxidant activity in plum wine, whereas *H. thilandica* SA1 had the opposite effect, with a slight decrease in TP content and antioxidant activity from the initial fermentation. However, the use of *H. thilandica* SA1 yeast produced more esters than other yeasts, resulting in a specific aroma to the plum wine. According to this study, plum can be processed into plum wine, which has better antioxidant activity compared with that of white wine. The study of wine fermentation using mixed cultures of these yeasts can provide guidance to future studies in terms of increasing the quality of the wine with respect to the TP content, antioxidant activity, and aroma.

## Figures and Tables

**Figure 1 molecules-28-03009-f001:**
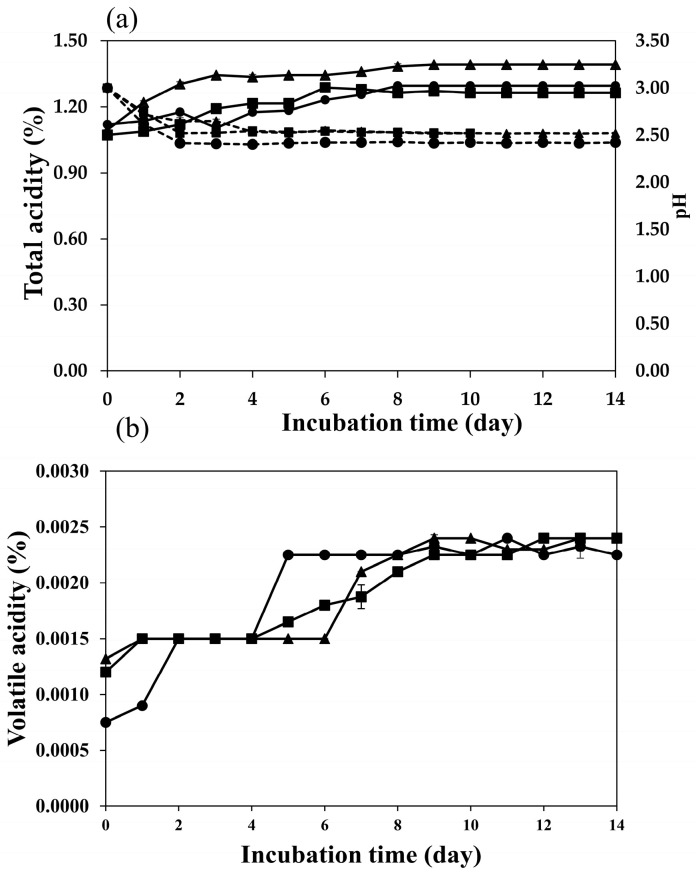
Total acidity (%) (solid line), pH (dash line), (**a**) and volatile acid content (**b**) of plum wine during wine fermentation using *S. cerevisiae* var. *burgundy* (●), *H. thailandica* Zal1 (■), and EC1118 (▲).

**Figure 2 molecules-28-03009-f002:**
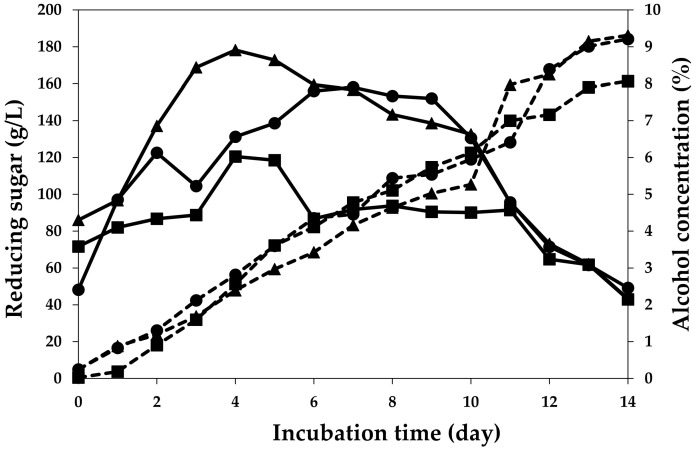
Reducing sugar (solid line) and alcohol contents (dash line) of plum wine during alcohol fermentation using *S. cerevisiae* var. *burgundy* (●), *H. thailandica* Zal1 (■), and EC1118 (▲).

**Figure 3 molecules-28-03009-f003:**
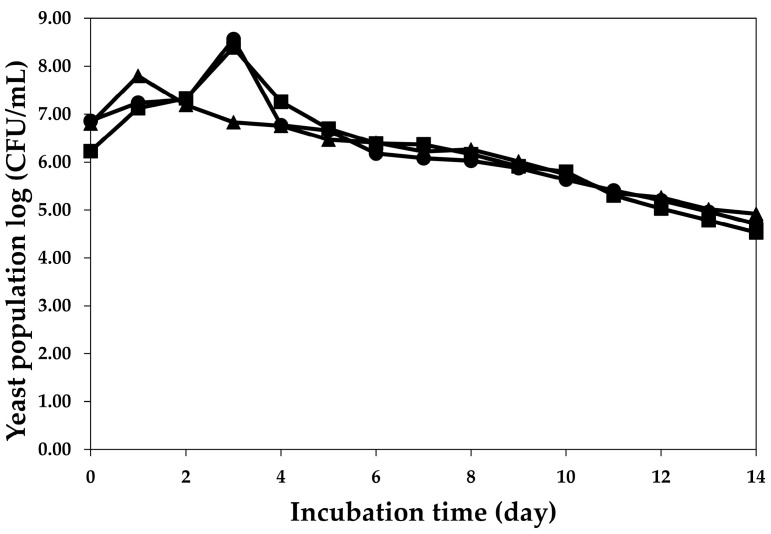
Change in yeast population during plum wine fermentation using *S. cerevisiae* var. *burgundy* (●), *H. thailandica* Zal1 (■), and EC1118 (▲).

**Figure 4 molecules-28-03009-f004:**
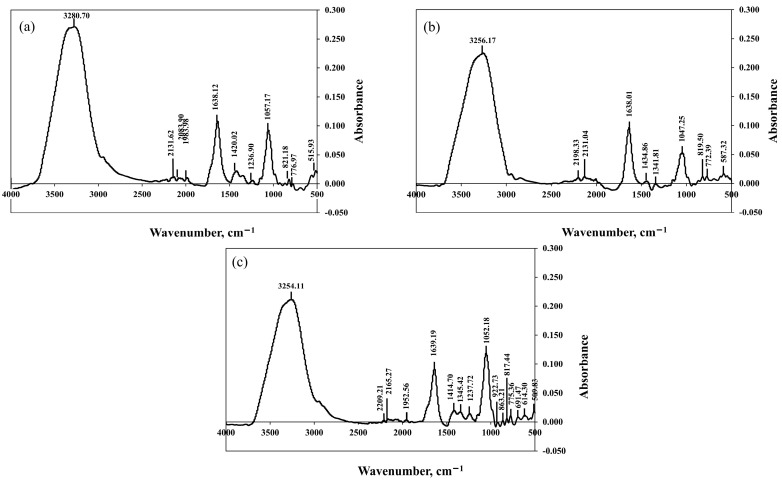
Fourier transform infrared (FTIR) spectra of plum wine using *S. cerevisiae* var. *burgundy* (**a**), *H. thailandica* Zal1 (**b**), and EC1118 (**c**).

**Table 1 molecules-28-03009-t001:** Total phenolic (TP) compounds and antioxidant activities of plum wines fermented by different yeast.

Yeast Species	TP (mg GAE/L)		DPPH (µg TE/g)	
	Day 0	Day 14	Day 0	Day 14
*S. cerevisiae* var *burgundy*	361.04 ± 75.32	469.84 ^a^ ± 6.95	272.50 ± 41.63	304.36 ^a^ ± 6.24
*H. thailandica* Zal1		331.48 ^c^ ± 16.24		241.65 ^c^ ± 0.94
EC1118		418.27 ^b^ ± 3.40		288.2 ^b^ ± 7.9

^a,b,c^ mean with the different letters in the same row of each kinetic parameter are significant at *p* ≤ 0.05.

**Table 2 molecules-28-03009-t002:** The volatile compounds identified in plum wine samples using gas chromatography-mass spectrometry (GC-MS).

Assignment Compounds	Retention Time			% Area			% Similarity		
	B	H	EC	B	H	EC	B	H	EC
Ethanol	1.287 ± 0.005	1.292 ± 0.002	1.292 ± 0.005	82.18 ± 2.19	71.97 ± 0.644	85.52 ± 0.300	95.7 ± 1.15	98.0 ± 0.0	95.0 ± 0.0
Acetic acid	1.407 ± 0.020	1.545 ± 0.000	-	0.47 ± 0.11	0.24 ± 0.000	-	90.0 ± 0.0	96.0 ± 0.0	-
Acetic acid ethyl ester	1.539 ± 0.002	1.545 ± 0.001	1.545 ± 0.000	8.51 ± 1.41	16.23 ± 0.008	7.37 ± 0.312	96.0 ± 0.0	96.0 ± 0.0	96.0 ± 0.0
Isobutylalcohol	1.585 ± 0.000	-	1.59 ± 0.000	0.35 ± 0.00	-	0.54 ± 0.087	96.0 ± 0.0	-	96.0 ± 0.0
Isopentyl alcohol	2.023 ± 0.001	2.029 ± 0.001	2.029 ± 0.000	8.81 ± 0.70	11.05 ± 0.036	6.58 ± 0.061	96.0 ± 0.0	96.0 ± 0.0	96.0 ± 0.0
Propanoic acid, 2-methyl-, ethyl ester	-	2.185 ± 0.000	-	-	0.37 ± 0.000	-	-	84.0 ± 0.0	-
2,3-Butanediol	2.334 ± 0.000	-	-	0.21 ± 0.00	-	-	96.0 ± 0.0	-	-
1-Butanol, 3-methyl-, acetate	-	3.535 ± 0.003	-	-	0.55 ± 0.015	-	-	97.0 ± 0.0	-

B = *S. cerevisiae* var. *burgundy*; H = *H. thailandica* Zal1; EC = EC1118.

## Data Availability

All data supporting the conclusions of this article are included in the manuscript.

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
