# Peer review of "Fermentation Characteristics and Aromatic Profiles of Plum Wines Produced with Hanseniaspora thailandica Zal1 and Common Wine Yeasts"

_molecules, 2023, doi:10.3390/molecules28073009_

Round 1
Reviewer 1 Report
The publication is interesting and well structured. The introduction is well structured and provides basic guidelines for the purpose of the study. Materials and methods are described in detail and correctly.
I have some notes on the paper.
1. The terms "wine", "wine fermentation" should be replaced or clarified by "plum wine fermentation", since only "wine" means that a drink is produced from grapes.
2. Is it correct to compare the fermentation data with the OIV standards which are for beverages derived from grapes?
3. All figures must be redone. They are currently too small and difficult to read.
4. At an appropriate place in the introduction, a comparison of the products obtained with Saccharomyces cerevisiae and the new types of wine obtained with Hanseniaspora thailandica should be made. This will enrich the publication and provide basic guidance as to why the research is being done.
Author Response
Response to Reviewer 1 Comments
The publication is interesting and well structured. The introduction is well structured and provides basic guidelines for the purpose of the study. Materials and methods are described in detail and correctly.
I have some notes on the paper.
Point 1: The terms "wine", "wine fermentation" should be replaced or clarified by "plum wine fermentation", since only "wine" means that a drink is produced from grapes.
Response 1: As your suggestion, it was corrected.
Point 2: Is it correct to compare the fermentation data with the OIV standards which are for beverages derived from grapes?
Response 2: To compare the effectiveness of data to white wine, solely to demonstrate that even though it is wine fermented from local fruits and is inexpensive, its properties such as antioxidant properties are not inferior to white wine fermented from grapes.
Point 3: All figures must be redone. They are currently too small and difficult to read.
Response 3: As your suggestion, all figures are uploaded in zip file.
Point 4: At an appropriate place in the introduction, a comparison of the products obtained with Saccharomyces cerevisiae and the new types of wine obtained with Hanseniaspora thailandica should be made. This will enrich the publication and provide basic guidance as to why the research is being done.
Response 4: As your suggestion, it was corrected in Page 2 ling 86-90
Reviewer 2 Report
The manuscript titled "Fermentation Characteristics and Aromatic Profiles of Plum Wines Produced with Hanseniaspora thailandica Zal1 and 3 common wine yeasts" tries to indicate possible new directions in plum processing. Although the combination of plum and different yeasts for winemaking is an interesting idea, it is not new. Furthermore, the experimental part is poorly explained, with unclear parts. Obtained results without statistical analysis, can be considered scientifically unreliable. The discussion is mainly descriptive but not profound. English is understandable but not good enough, so language correction is mandatory
Bellow were listed some points that must be clarified or modified.
Lines 54-65. Reading the mentioned literature, I understood that fermentation is carried out. First sentences in literature you cited (Fine wine master, 2022): "As the name suggests, plum wine is traditionally made from Japanese Ume plums fermented in sugar and yeast.”
Apart from that, it is not very clear what the authors mean by liquor. Do you mean a type of alcoholic drink or an ethanol solution? Please, revised this paragraph.
Line 80. Correct the word “fucus”
Line 83. Saccharomyces should be in italic
Line 99. Duplication of results. Table 1. should be deleted because these results are already mentioned in the text.
Figure 1a. Undefined right axes.
Line 134. Please, provide an explanation why the fermentations were stopped after 14 days. What about the remaining sugar? There was enough sugar for yeasts (especially EC1118 and S. cerevisiae var. burgundy) to continue the fermentation.
Figure 3. The left axes is the log CFU/ml
Line 213. The results should be presented as mean values ± standard deviations
Line 272. Specify the quantity of fermentation mixture per sample (total volume of plum, sugar and water inoculated and subjected to the fermentation).
Lines 274-275. Please, describe inoculation with starter culture in more detail.
If the authors inoculated 5% of the total volume (total mixture volume of plum, sugar and water), what was the initial CFU per ml? In the Figure 3 the initial CFU for the H. thailandica SA1 was higher than 106 CFU/ml, although it is stated in the experimental part that 105 CFU/mL was used (line 264).
On the other hand, how the authors confirmed that this cell number is enough to ensure the development of inoculated strains capable to cause a substantial impact on wine quality. What was the exact quantity of EC1118, 0.25 or 0.4 g/L? In order to obtain comparable results, the inoculated yeast cell numbers should be the same for the different yeasts.
Line 293. To avoid ambiguity, I think that is a good idea to combine parts 3.2. and 3.4. in one.
Experimental part. In the experimental part it is stated that the measurements were repeated three times and that the results were subjected to statistical analysis. Please add the obtained results in the text and indicate in tables if there were significant differences between the mean values of the results.
Author Response
Response to Reviewer 2 Comments
The manuscript titled "Fermentation Characteristics and Aromatic Profiles of Plum Wines Produced with Hanseniaspora thailandica Zal1 and 3 common wine yeasts" tries to indicate possible new directions in plum processing. Although the combination of plum and different yeasts for winemaking is an interesting idea, it is not new. Furthermore, the experimental part is poorly explained, with unclear parts. Obtained results without statistical analysis, can be considered scientifically unreliable. The discussion is mainly descriptive but not profound. English is understandable but not good enough, so language correction is mandatory
"We use grammar verification to ensure correct writing as recommended by the committee."
Bellow were listed some points that must be clarified or modified.
Point 1: Lines 54-65. Reading the mentioned literature, I understood that fermentation is carried out. First sentences in literature you cited (Fine wine master, 2022): "As the name suggests, plum wine is traditionally made from Japanese Ume plums fermented in sugar and yeast.”
Apart from that, it is not very clear what the authors mean by liquor. Do you mean a type of alcoholic drink or an ethanol solution? Please, revised this paragraph.
Response 1: As your suggestion, it was corrected in Page 2 ling 56-59.
Point 2: Line 80. Correct the word “fucus”
Response 2: As your suggestion, it was corrected in Page 2 ling 87.
Point 3: Line 83. Saccharomyces should be in italic
Response 3: As your suggestion, it was corrected in Page 2 ling 90.
Point 4: Line 99. Duplication of results. Table 1. should be deleted because these results are already mentioned in the text. Figure 1a. Undefined right axes.
Response 4: As your suggestion, Table 1 was deleted and Figure 1a was corrected.
Point 5: Line 134. Please, provide an explanation why the fermentations were stopped after 14 days. What about the remaining sugar? There was enough sugar for yeasts (especially EC1118 and S. cerevisiae var. burgundy) to continue the fermentation.
Figure 3. The left axes is the log CFU/ml
Response 5: In this research, the fermentation process lasted for 14 days due to the trend of beverage consumption in Thailand, which was also influenced by traditional umeshu drinkers. Therefore, the usual fermentation time for wine was not used in order to achieve the appropriate alcohol content while retaining some residual sugar for a sweet taste, similar to that of a sweet wine.
Figure 3 was corrected.
Point 6: Line 213. The results should be presented as mean values ± standard deviations
Response 6: As your suggestion, it was corrected in Table 2.
Point 7: Line 272. Specify the quantity of fermentation mixture per sample (total volume of plum, sugar and water inoculated and subjected to the fermentation).
Response 7: As your suggestion, it was corrected in Page 9 line 280-281. Regarding the sugar content, we cannot determine the exact amount as it depends on the Brix of the mixture at that particular time.
Point 8: Lines 274-275. Please, describe inoculation with starter culture in more detail.
If the authors inoculated 5% of the total volume (total mixture volume of plum, sugar and water), what was the initial CFU per ml? In the Figure 3 the initial CFU for the H. thailandica SA1 was higher than 106 CFU/ml, although it is stated in the experimental part that 105 CFU/mL was used (line 264).
On the other hand, how the authors confirmed that this cell number is enough to ensure the development of inoculated strains capable to cause a substantial impact on wine quality. What was the exact quantity of EC1118, 0.25 or 0.4 g/L? In order to obtain comparable results, the inoculated yeast cell numbers should be the same for the different yeasts.
Response 8: As your suggestion, it was corrected in Page 9 line 273-274 and 285. The initial CFU was 106 CFU/ml. For substantial impact on wine quality, we cited from Chanprasartsuk et al, 2012 with initial 106 CFU/ml for stater.
Point 9: Line 293. To avoid ambiguity, I think that is a good idea to combine parts 3.2. and 3.4. in one. Experimental part. In the experimental part it is stated that the measurements were repeated three times and that the results were subjected to statistical analysis. Please add the obtained results in the text and indicate in tables if there were significant differences between the mean values of the results.
Response 9: As your suggestion, parts 3.2 and 3.4 were combined. We also added the SD value in Table 2 and added the text in Page 6 line 180-181.

Round 2
Reviewer 2 Report
The authors have provided a nicely detailed and thorough response to the comments from the previous review and have addressed my major concerns.
I do not have any further comments.